# Multiscale Cascaded Attention Network for Saliency Detection Based on ResNet

**DOI:** 10.3390/s22249950

**Published:** 2022-12-16

**Authors:** Muwei Jian, Haodong Jin, Xiangyu Liu, Linsong Zhang

**Affiliations:** 1School of Computer Science and Technology, Shandong University of Finance and Economics, Jinan 250014, China; 2School of Information Science and Technology, Linyi University, Linyi 276012, China

**Keywords:** ResNet, multiscale cascade extraction module, attention module, saliency detection

## Abstract

Saliency detection is a key research topic in the field of computer vision. Humans can be accurately and quickly mesmerized by an area of interest in complex and changing scenes through the visual perception area of the brain. Although existing saliency-detection methods can achieve competent performance, they have deficiencies such as unclear margins of salient objects and the interference of background information on the saliency map. In this study, to improve the defects during saliency detection, a multiscale cascaded attention network was designed based on ResNet34. Different from the typical U-shaped encoding–decoding architecture, we devised a contextual feature extraction module to enhance the advanced semantic feature extraction. Specifically, a multiscale cascade block (MCB) and a lightweight channel attention (CA) module were added between the encoding and decoding networks for optimization. To address the blur edge issue, which is neglected by many previous approaches, we adopted the edge thinning module to carry out a deeper edge-thinning process on the output layer image. The experimental results illustrate that this method can achieve competitive saliency-detection performance, and the accuracy and recall rate are improved compared with those of other representative methods.

## 1. Introduction

The task of visual saliency detection was created to allow computer systems to mimic the capabilities of the human visual system (HVS) for quickly extracting salient objects from a scene. These saliency regions in an image/video usually contain the object of interest to the observer and those areas that can gain HVS attention in real life. With the in-depth study of convolutional neural networks, saliency detection has been widely applied as an effective technique for preprocessing numerous content-based tasks in computer vision, such as image recognition, image segmentation, image retrieval, and pedestrian/object detection [1,2,3].

Early vision work was classified based on viewpoint acquisition mechanisms into cognitive [4,5,6,7,8], Bayesian [9], spectral analysis [10], information-theoretic [11], graphical [12], decision-theoretic [13], and pattern classification models [14]. With the progress of saliency detection, image-oriented detection methods have formed more complete detection systems, which can be separated into two main groups. One is the task-driven top-down detection methods, which often require a training process of task-dependent and specific prior knowledge; the other one is data-driven and subconscious, bottom-up detection methods, which mainly use underlying visual cues such as color, contrast, and shape for saliency object. In addition, in pace with the advancement and development of imaging devices, depth information acquisition is becoming easier and more convenient to manipulate, which has created the groundwork for the rise and progress of RGBD image saliency-detection algorithms [15]. Compared with research on traditional 2D image saliency detection [16,17], the research on RGBD image saliency-detection algorithm started late and has achieved certain satisfactory results. However, researchers have not reached a consensus on the mechanism through which the effect of depth information on the human perceptual system is achieved and how to effectively explore depth information; thus, further in-depth research is still needed.

Even though many of saliency-detection methods have achieved notable results, they are still not satisfactory in removing background interference, maintaining unabridged edges, and other slight details. To address the shortcomings of these conventional methods, we developed an image saliency-detection network using the classical convolutional neural network model as the basic framework, and designed an efficient saliency-detection model based on a multiscale cascaded attention network. In summary, the main contributions of this study are characterized as follows:(1)We employ a multiscale cascade block and a lightweight channel attention module between the typical encoding–decoding networks for optimizing the performance of image saliency detection based on ResNet34.(2)A multiscale cascaded attention model is devised to rationally use the multiscale extraction module for high-level semantic features of the image, while the attention module is used for the joint refinement of low- and high-level semantic features to enhance the precision of saliency detection.(3)To solve the problem of blurred edges that has been neglected in many existing methods, we applied the edge refinement module to the output layer image for clear edge refinement.

The remainder of this paper is structured as follows: We first describe the present status of the associated work in Section 2. The designed network architecture and loss function are outlined in Section 3. Additionally, Section 4 provides the outcomes of our experiments. Finally, Section 5 presents our conclusions and discussion.

## 2. Related Work

### 2.1. Traditional Saliency Detection Methods

Traditional saliency-detection methods can be coarsely separated into spatial-domain-based and transform domain-based modeling frameworks. Spatial-domain-based detection approaches are usually studied based on image processing theory, with the output of saliency detection being results generated by low-level cues (contrast, chrominance, luminance, texture, etc.). These methods usually perform pixel-level saliency region extraction by calculating the difference between the pixels in the salient region and the surrounding background pixels; thus, they depend on the size of the selected window and the threshold value for saliency discrimination. A typical strategy based on low-level features is to extract the salient regions by optimizing thresholds. The AC algorithm [18] is a numerically computed saliency mapping generation algorithm, where the local comparison between the input image’s area R1 and its neighborhood R2 at various scales determines the saliency value. Later, with the increases in data volume and the accuracy requirement on extracted images, the optimized thresholding approach was replaced by other advanced methods due to its limitation of not being applicable in many images with complex textures. Cheng et al. [19] created different approaches combined with global contrast, named histogram-based contrast (HC) and regional contrast (RC) salient object models.

In recent years, spatial-domain-based methods are more often implemented based on image component analysis strategy, the core tenet of which is analyzing the principal component and independent component, and using other spatial variation methods to explore the correlation between image foreground and background pixels to achieve salient region extraction. For instance, Goferman et al. [20] devised a saliency method focused on context, which can detect the salient regions in representative scenes rather than just salient objects. Additionally, in the spatial domain, the graph-theory-based saliency-detection model usually splits the inputs into diverse blocks and regards each of them as nodes. Then, weighted edges between blocks of pixels, depending on visual characteristics such as color, luminance, and orientation, are integrated to determine the graphical mapping [21]. Harel et al. [12] designed a method named graph-theory-based algorithm, which simulates the visualization principle in the feature extraction process. Specifically, in the stage of generating the saliency map, Markov chains are introduced, and the central surround difference is calculated with a graph model. The saliency map is then obtained by a purely mathematical calculation.

The spatial-domain-based methods can achieve satisfactory results on images with certain differences between the foreground and background, but the results are not ideal for many images without significant differences in the spatial domain. In order to tackle the limitations of these spatial domain methods, many transform-domain methods based on Fourier transform and wavelet/Gabor transform have been exploited in the field of saliency detection. Transform-domain-based methods generally include wavelet transform [22], wavelet frame transform [23], curvelet transform, projection transform, etc. Although transform operators such as Fourier transform can more accurately describe the global and macroscopic features, the results of this method are not acceptable for local or unsmooth information. Hou and Zhang [10] developed a spectral residual algorithm, which is a typical task in the field of frequency-domain-based saliency detection. In detail, this strategy considers the possibility of distributing an image’s substantial content as salient and redundant information. The log spectrum distribution exhibits a consistent trend for various data, and the curve complies with the local linearity requirement.

Based on previous studies, Guo et al. [24] exploited a novel method, the phase spectrum of quaternion Fourier transform (PQFT), which abandons the magnitude spectrum and only utilizes the input image’s phase spectrum following Fourier transformation. Saliency mapping similar to that of the SR method is obtained by Fourier inverse transform. The Fourier-transformed phase spectrum is expanded. After that, Achanta et al. [25] furthered the FT algorithm and devised the maximum symmetric surround method for saliency detection. This method varies the center surrounding bandwidth according to the separation between a pixel’s point and the edge of an image. Thus, the algorithm uses the average of the most likely symmetric neighboring areas rather than calculating the average of the global feature vectors generated by the FT method. Although traditional techniques have taken some steps in the domain of saliency detection, they still cannot adapt to the numerous high-complexity and low-quality data.

### 2.2. Deep-Learning-Based Saliency-Detection Methods

As it is difficult to improve the effect of traditional saliency detection methods, deep learning-based approaches have received the attention of scholars in recent years. Since the earliest BP networks [26], a group of saliency detection frameworks have appeared in the field of machine learning. After the accelerated advancement of neural networks [27,28,29,30], more models based deep learning have emerged. Since 2015, saliency detection has been processed by convolutional neural networks (CNNs). Unlike the traditional techniques based on comparison of visual cues, CNN-based methods effectively reduce the need to design manual features and greatly improve the computing efficiency, so these methods have been extensively used by many scientific scholars [31,32,33].

CNN-based models typically contain many neurons with adjustable parameters and variable structural field sizes. The neurons have a large receptive range to provide global information, which makes it possible to identify the regions of salient objects in the scenario more effectively. Compared with traditional methods, CNN and its optimization methods have become the most mainstream methods at the current research stage due to their excellent extraction accuracy and computational efficiency. Wang et al. [34] suggested a visual attention module based on global saliency feature information for visual saliency-detection networks, which focuses on both superficial refined layers with locally salient responses and deep coarse layers with globally salient information. Cornia et al. [35] designed an architecture that incorporates neural attention mechanisms to generate saliency maps. Zhu et al. [36] designed a multiscale adversarial feature learning (MAFL) model for saliency detection. Recently, Wei et al. [37] introduced a deep saliency-detection framework using full convolutional networks (FCNs) to solve the cosalient object discovery and detection problem. He et al. [38] devised a new superpixel-based framework called SuperCNN, which can better extract the interior representations of saliency and hierarchical contrast features independent of the region size by using a multiscale network structure. Later, Hou et al. [39] designed a new saliency-detection method stemming from holistically nested edge detection (HED) by adding a skip layer structure, where high-level features guide low-level features, thus forming an efficient end-to-end salient-object-detection method. Hui et al. [40] exploited a multiguided saliency-detection model using the intrinsic relationship between different features. To further improve the performance and robustness, a novel pixel-by-pixel contrast loss function was developed and integrated with the cross-entropy loss function to jointly supervise the training process. Recently, as a key advance in deep learning, a transformer-based network is applied to salient object detection. Liu et al. [41] introduced a pure transformer into saliency detection to make a convolution-free model called visual saliency transformer (VST). For better extract low- and high-level information, Hussain et al. [42] designed a parallel architecture to integrate both transformer and CNN features, which are fed into a pyramidal attention module.

## 3. Proposed Method

Regarding the problems commonly experienced in the current research, in this section, we propose a multiscale cascaded attention network for salient object detection. Firstly, the image data et is preprocessed in real time; i.e., after the object is locked, grayscale images matching the region are generated and fused, so as to effectively eliminate the background noises and improve the accuracy of the salient region detection. The devised model is able to reduce the interference of redundant objects by accordingly processing the multiobject images. Then, the preprocessed images are put into the multiscale cascaded attention network for saliency detection. The extraction part of the consists of an encoder, extraction block, and composition, in which the extraction of low-level features by the encoder and the extraction of high-level semantic features by the multiscale cascade-attention module are jointly utilized to enhance the performance of saliency detection for the whole and detailed parts. Finally, the extracted information is integrated using the decoder network to obtain the final saliency-detection map. We compared our method with nine advanced methods on three public datasets (DUTS [43], ECSSD [44], and HKU-IS [45]), and the experiments demonstrated that this method is advantageous in terms of overall metrics and visual details. The general structure of the network is shown in Figure 1.

### 3.1. Network Architecture

The overall framework of the proposed method is presented in Figure 1. The input image is preprocessed with the YoLoV3 network [46] to eliminate most of the interference from the background, the processed image is fed into the U-shaped backbone network for feature extraction and processing, and then the extracted feature map is upsampled to generate the final saliency image. The following subsections provide a detailed description of the entire network.

#### 3.1.1. Object Locking and Extraction from Images

In this subsection, the preprocessing stage to highlight the salient object as well as to remove the background interference is systematically introduced. During image preprocessing, the initial image is directly input to the preprocessing module, and then the processed intermediate image is used as the input of the saliency extraction network for subsequent processing. With regard to object tracking, Redmon et al. [46] improved the YoLo network and designed a model that assigns only one bounding box to each object. Compared with C-RNN, this network significantly improves the processing speed of the YoLoV3 network because the coordinates of the bounding box are directly predicted and localized using the convolutional extraction of the features, followed by the fully connected layer. The YoLoV3 network is a further improvement of the YoLo series of algorithms, which retains the advantages of the previous algorithms while improving the accuracy. YoLoV3 has strengthened performance and increased speed, so has become one of the preferred detection algorithms in the engineering community due to its powerful real-time performance and concise network structure. In applications, the coordinates of the center of the selected object can be obtained through the YoLoV3 network, and the generated center coordinates are used to produce a matching grayscale image, which is then fused with the original input image. For the natural images in the test set, the relative position and area information of the detected objects are accurately acquired during the preprocessing stage. After inputting an image to the module, it is first sent to the preprocessing module, and the specific formula for determining the bounding box is formulated as follows:(1)$$b_{x}=\sigma \left(t_{x}\right)+C_{x},$$
(2)$$b_{y}=\sigma \left(t_{y}\right)+C_{y},$$
(3)$$b_{w}=P_{w}e^{t_{w}},$$
(4)$$b_{h}=P_{h}e^{t_{h}},$$
where $t_{x}$, $t_{y}$, $t_{w}$, and $t_{h}$ denote the four coordinates used to predict the object bounding box in the YoLo network; $C_{x}$ and $C_{y}$ represent the horizontal and vertical offsets of the network where the center of the object is located and the coordinates of the upper left corner of the object image, respectively; and $P_{w}$ and $P_{h}$ indicate the width and height of the corresponding bounding box, respectively.

After we obtain the parameters of the object bounding box, we generate the corresponding grayscale image based on the extracted coordinate information of the object and merge the grayscale image with the original image to filter out the purposeless background information. Regarding the visual perception mechanism mentioned above, when the observer extracts the object of interest in the scene, if the observer closely watches the object, the gaze distance decreases, and the field of view becomes smaller, but the object will be clearer. On the contrary, as the gaze distance increases, the observer’s field of view gradually increases, but the clarity of the salient object becomes increasingly blurred. At the same time, if the gaze distance remains the same, the objects around the region of interest decrease in sharpness as the distance from the central object increases. Based on this principle, we generate a grayscale image matching the coordinate of the object frame, and add the α channel, which denotes transparency. The value of α is calculated by the Gaussian function, as shown in the following calculation formula:(5)$$\alpha (x_{i},\;y_{i})=e\frac{\sqrt{{\left(x_{c}-x_{i}\right)}^{2}+{\left(y_{c}-y_{i}\right)}^{2}}}{s\times s},$$
where $\left(x_{c},\;y_{c}\right)$ are the horizontal and vertical coordinates of the center of the object in the bounding box, respectively; $\left(x_{i},\;y_{i}\right)$ represent the position of the corresponding pixel in the image; and the value of s is dynamically set according to the width and height of the bounding box. Thereafter, α can be further estimated. When the pixel in the generated grayscale image lies within the object bounding box $Area_{b}$, the corresponding value is the actual value of the pixel, and the value of α for that point is constant. When the pixel of the grayscale image is not located in the boxed area, the value is set to 255. The final preprocessed image resulting from the fusion of the grayscale image with the input image is expressed with the following equation:(6)$$\alpha (x_{i},\;y_{i})=\left\{\begin{array}{cc} \alpha , & (x_{i},\;y_{i})\in Area_{b}\\ 255, & others \end{array}\right.,$$

Some samples of the preprocessed images are displayed in Figure 2. Occasionally, multiple objects may occur during the process. In this situation, we detect the horizontal and vertical coordinates of each coordinate point of all objects in an image, and select the largest and smallest horizontal and vertical coordinates to determine the two coordinate points. These two coordinate points form a bounding box, which can also effectively remove other distracting backgrounds.

#### 3.1.2. Feature Encoder Module

The core framework of the network devised for saliency detection in this study includes three main modules: the feature encoder module, the contextual feature extraction module, and the feature decoder module. In this subsection, the feature encoder is introduced. As shown in Figure 3, based on the design of U-Net, the encoder is replaced with a pretrained ResNet34. The designed structure preserves the feature extraction modules and the average pooling layer, but discards the final fully connected layer of the origin network. For better expression, the dimensional transformation of the feature map is clearly represented.

#### 3.1.3. Contextual Feature Extraction Module

This subsection focuses on the second part of the network structure—the contextual feature extraction module. This module consists of two distinct parts. One is a multiscale cascade block to perform multiscale feature extraction, while the other part is a lightweight channel attention module to perform feature refinement. This entire module is intended to enhance the semantic information of the context and allows the generation of higher-level feature maps.
(1)Dilated Convolution

Deep convolutional layers have been shown to be an efficient for generating visual feature representations for tasks such as semantic segmentation and object detection. However, the pooling layer may cause the original image’s semantic information to be lost. In order to overcome this limitation, we employ dilated convolution for this process to enhance the efficiency of computation, and this operation is formulated as below:(7)$$y(i)=\sum_{k}x(i+rk)w(k),$$
where x represents the input feature map, y expresses the output feature map, w indicates the filter, and r denotes the dilation rate when sampling the image. Typically, the standard convolution is a special case of r=1. In contrast, in the multiscale extraction module, the dilated convolution allows us to change this rate to adaptively modify the receptive field of the filter. This process is illustrated in Figure 4.
(2)Multiscale Cascade Block (MCB)

Both the inception structure and ResNet network are typical and representative frameworks based on deep learning. The inception structure widens the network architecture using various receptive fields, whereas ResNet uses a skip connection method to prevent gradients from explosion and disappearance. Therefore, the multiscale cascade block applies the inception structure to splice with the decoder’s ResNet network as a way of inheriting the advantages of both.

As shown in Figure 5, the proposed multiscale cascade block has four cascade branches. The convolutions with various dilation rates are sequentially added from top to bottom: each branch contains 3, 7, 9, and 19 perceptual fields, respectively. We use a 1 × 1 convolution for the activation of every branch. Afterward, the original features and other multiscale features are summed. In this module, the convolution with a larger receptive field is extracted for larger objects and generates more contracted features. The convolution of smaller receptive fields achieves better results for the extraction of small objects. Additionally, through combining the convolution with various dilation rates, the multiscale cascade block can simultaneously extract the salient features of diverse objects of different sizes.
(3)Channel Attention (CA) Module

To further strengthen the accuracy of the saliency detection results, we also added a lightweight channel attention module for saliency detection optimization through assigning greater weights to the high response channels.

#### 3.1.4. Feature Decoder Module

In this study, we exploited the feature decoder module to recover the high-level semantic features obtained by the feature encoder and the contextual feature extraction modules. Our decoding network is almost symmetrical with the first half of the encoder. Each stage in the decoder includes three convolutional layers with normalization and ReLU activation functions. The input of each stage is a feature mapping of the connection between its previous stage and its upsampled output of the corresponding stage in the encoder, and then the multichannel output of each decoder is fed into a 3 × 3 convolutional layer, followed by a bilinear upsampling and a sigmoid function. Moreover, there is a separate supervision in each stage as a means of achieving intensive deep supervision during the training process and increasing the accuracy of the saliency-detection results. The structure of the decoder is illustrated in Figure 6.

### 3.2. Loss Function

During the training process of the framework described above, the computed saliency map with the labeled dataset is learned by the loss estimation between them [35,47]. In binary classification studies, the binary cross-entropy (BCE) loss function has been frequently used. It also an objective function typically used to measure the difference between the predicted saliency maps and ground truth in saliency detection tasks, which has widely achieved good results. Therefore, a BCE loss function was employed in this study. The expression of binary cross-entropy loss is formulated as:(8)$$L(\theta ,w)=l_{predict}(\theta ,w_{predict}),$$
where θ represents the set of all network parameters, w indicates the weight of the corresponding layer, and l is the binary cross-entropy loss function, which can be employed to equalize the generated saliency value $Y\in {\left(0,1\right)}^{N}$ and its corresponding labeled image $G\in {\left(0,1\right)}^{N}$ as follows:(9)$$L=-\sum_{i=1}^{N}\left\{\left(1-a)g_{i}\log y_{i}+a(1-g_{i})\log(1-y_{i}\right)\right\}$$
where N=H×W represents the image size, $g_{i}\in G$ and $y_{i}\in Y$, accordingly.

## 4. Experiments

### 4.1. Implementation Details

In this study, the deep-learning framework for the experiments was built on Pytorch, and other detailed environment configurations are indicated in Table 1.

### 4.2. Qualitative Analysis

In this part of the study, visual comparisons of saliency-detection results were evaluated. To validate the effectiveness of the devised saliency-detection network, we selected nine existing saliency-detection methods for contradistinctive experiments, namely Amulet [8], DCL [9], DHS [10], MDF [11], NLDF [12], UCF [13], RAS [14], R3Net [48], and DGRL [49]. Three different datasets, including DUTS [43], ECSSD [44], and HKU-IS [45], were tested. In Figure 7, the first column represents the example image, the second column expresses the ground truth of the highlighted salient objects, and the third column displays the saliency map produced by the proposed method. The subsequent columns are the saliency results for each different method. The probability that each pixel point in the image belongs to the foreground is represented its corresponding pixel value. It can be seen from Figure 7 that our devised model produced saliency results nearer to the ground truth.

In Figure 7, the first three rows are salient object detection comparisons on the DUTS dataset; the fourth and fifth rows are detection results in terms of the ECSSD dataset and the HKU-IS dataset, respectively. It can be distinctly observed that the results obtained by the proposed method in the first and third rows are more accurate than those of the comparative methods in the overall extraction of the deer and human. Moreover, the experimental data in the third row show that our method also obtained better result in the details, such as the tentacles of flying insects and the tiny legs, compared with the other nine approaches used for comparison. In addition, the fifth row reflects that our method was not only effective in extracting single objects, but could also maintain efficient saliency detection for multiple objects. In summary, the proposed method is more effective than the other nine comparative approaches in terms of visual comparison, and our results resembled the ground truth. The results of the comparison indicated that the devised model is able to generate more accurate salient maps.

### 4.3. Quantitative Analysis

Next, we quantitatively analyzed the experimental results by comparing evaluation metrics. The results of the precision–recall (P-R) curves of each model on three distinct image datasets are presented in Figure 8. Specifically, Figure 8a displays the P-R curves of the individual algorithm on the DUTS dataset, Figure 8b reveals the P-R curves of each model on the ECSSD dataset, and Figure 8c shows the P-R curves of diverse approaches on the HKU-IS dataset. Figure 8 shows that the proposed method has a greatly improved accuracy and recall rate compared with those of the other methods.

Additionally, Table 2 presents the $F^{\max}$ (*F* measure) evaluation metric, $F^{\omega }$ (weighted *F*-measure score) and mean absolute error (*MAE*) score of the different methods on the three datasets. Table 2 shows that the proposed method has an improved $F^{\max}$ on all of three datasets compared with those of the other nine methods. Our method was able to achieve $F^{\max}$ values of 0.832, 0.932, and 0.917 and $F^{\omega }$ scores of 0.736, 0.865, and 0.844 on DUTS, ECSSD, and HKU-IS, respectively.

MAE can reflect the accuracy of the model in terms of the error rate of the detection results, and Table 2 shows that our method obtains the smallest MAE values among all the methods, which were reduced to 0.052, 0.041, and 0.035 on the three data sets. Although one of the methods used for comparison achieved the same value as our method, this is enough to reflect the devised model is capable of achieving promising results. The results of the many experiments indicated that the designed image saliency-detection framework is feasible and effective.

## 5. Conclusions

In this study, a multiscale cascaded-attention framework was developed for saliency detection, which overcomes the shortcomings of existing methods, such as the edges of salient objects being not clear enough and the presence of background interfering with the saliency map. The main network framework of the proposed method was inspired by U-Net, while the ResNet was designed as a U-shaped network for saliency detection optimization. Specifically, a multiscale cascade block (MCB) and a lightweight channel attention (CA) module were jointly added between the encoding and decoding networks for optimization. Eventually, the visual attention mechanism was exploited for feature extraction, and integration was performed to refine the saliency-detection results. The experimental results illustrated that the designed method produces competitive saliency-detection performance and has higher accuracy and recall than other methods. Recently, as a key advance in deep learning, the transformer-based network has also started to be applied to salient object detection. Limited by the experimental equipment, our framework was designed based on the convolutional neural network. Therefore, exploring transformer-based saliency methods is our future research direction.

## Figures and Tables

**Figure 1 sensors-22-09950-f001:**
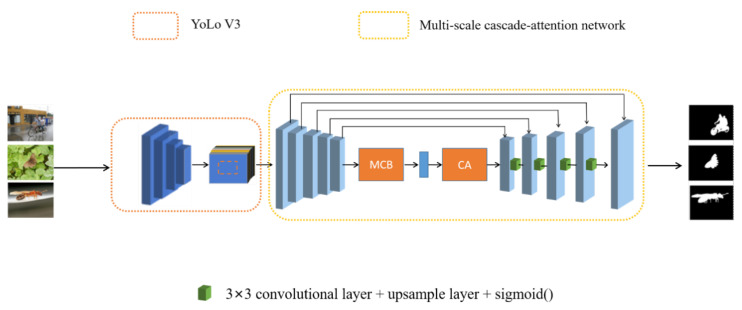
Overall architecture of the proposed network.

**Figure 2 sensors-22-09950-f002:**
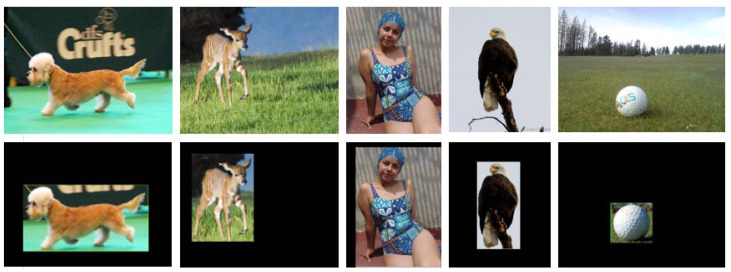
Some sample images with salient objects after preprocessing module.

**Figure 3 sensors-22-09950-f003:**
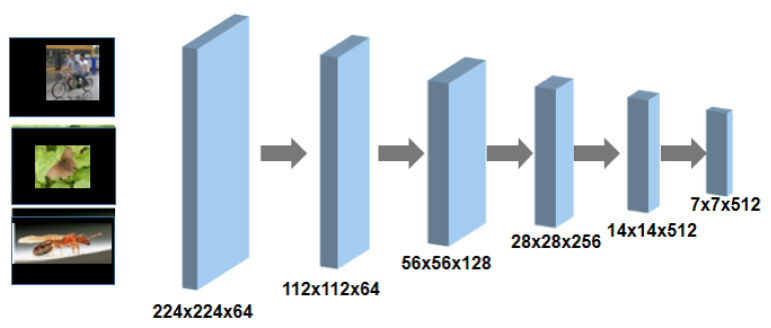
The designed architecture of the feature encoder module.

**Figure 4 sensors-22-09950-f004:**
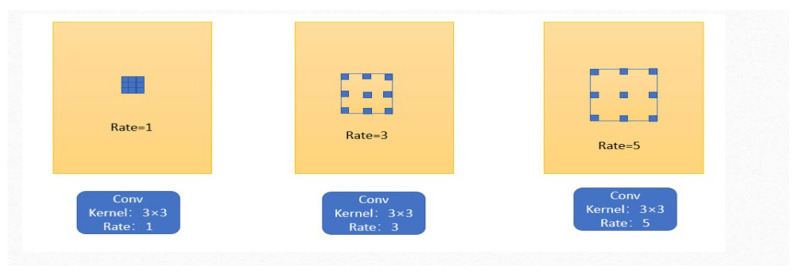
Schematic diagram of dilated convolution.

**Figure 5 sensors-22-09950-f005:**
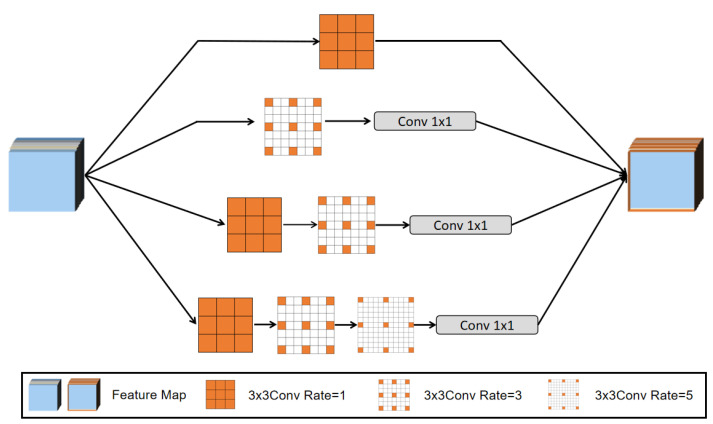
The devised architecture of multiscale cascade block.

**Figure 6 sensors-22-09950-f006:**
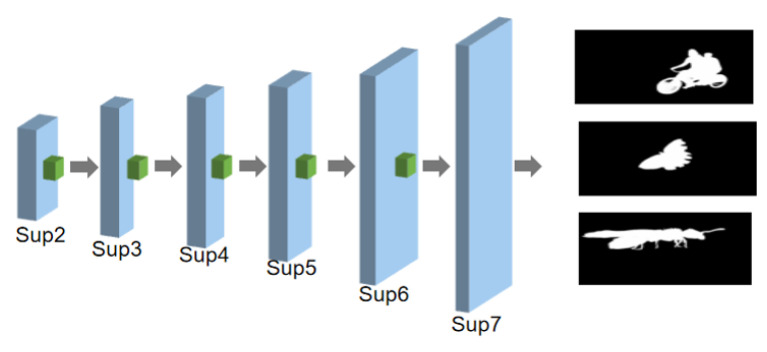
The devised architecture of the feature decoder module.

**Figure 7 sensors-22-09950-f007:**
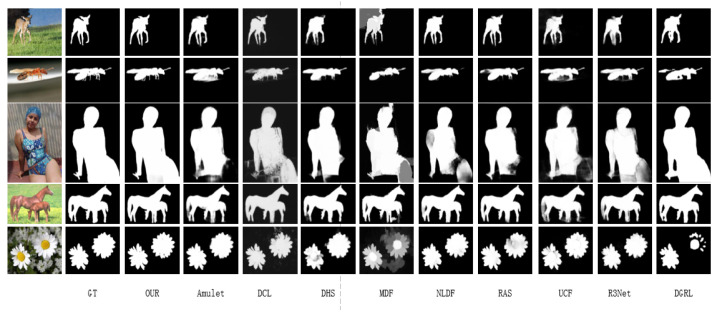
The visual comparison of each model on different datasets.

**Figure 8 sensors-22-09950-f008:**
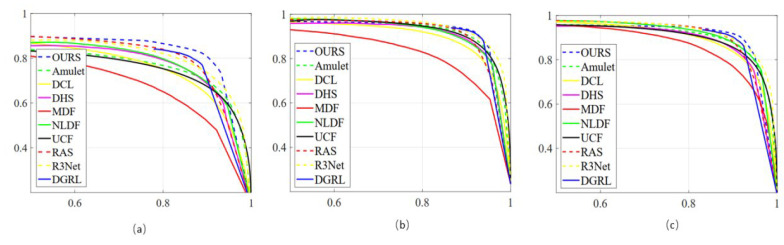
The P-R curves of each method according to different datasets. (**a**) the P-R curves of the individual algorithm on the DUTS dataset; (**b**) the P-R curves of each model on the ECSSD dataset; (**c**) the P-R curves of diverse approaches on the HKU-IS dataset.

**Table 1 sensors-22-09950-t001:** Configuration details of the experimental implementation.

Experimental Implementation	Configuration
Operating System	Win10
Python	3.7
Pytorch	1.5.0
CUDA	9.0
GPU	NVIDIA-GTX1080ti

**Table 2 sensors-22-09950-t002:** Comparison of the *F*-measure evaluation metric and MAE score on different dataset.

Methods	DUTS			ECSSD			HKU-IS		
	$F^{\max}↑$	$F^{\omega }↑$	MAE↓	$F^{\max}↑$	$F^{\omega }↑$	MAE↓	$F^{\max}↑$	$F^{\omega }↑$	MAE↓
Amulet [8]	0.778	0.657	0.085	0.915	0.841	0.059	0.895	0.813	0.052
DCL [9]	0.782	0.606	0.088	0.890	0.802	0.088	0.885	0.736	0.072
DHS [10]	0.807	0.698	0.067	0.832	0.841	0.059	0.890	0.806	0.053
MDF [11]	0.730	0.509	0.094	0.783	0.605	0.105	0.861	0.726	0.129
NLDF [12]	0.812	0.710	0.066	0.905	0.839	0.063	0.902	0.838	0.045
UCF [13]	0.771	0.588	0.117	0.911	0.789	0.078	0.886	0.751	0.074
RAS [14]	0.831	0.727	0.060	0.920	0.809	0.056	0.913	0.821	0.045
R3Net [48]	0.828	0.715	0.059	0.931	0.832	0.046	0.916	0.837	0.038
DGRL [49]	0.829	0.708	0.050	0.922	0.813	0.041	0.910	0.842	0.036
DSS [16]	0.825	0.732	0.057	0.915	0.858	0.052	0.913	0.836	0.039
PiCANet [50]	0.851	0.748	0.054	0.931	0.863	0.042	0.921	0.847	0.042
CSNet [51]	0.819	0.712	0.074	0.916	0.837	0.066	0.899	0.813	0.059
Ours	0.832	0.736	0.052	0.932	0.865	0.041	0.917	0.844	0.035

## Data Availability

Not applicable.

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
