# Peer review of "Multiscale Cascaded Attention Network for Saliency Detection Based on ResNet"

_sensors, 2022, doi:10.3390/s22249950_

Round 1
Reviewer 1 Report
The authors introduce the essence of salient object detection and propose a new framework to improve the defects during saliency detection based on ResNet34 network. They also devise a contextual feature extraction module to enhance the advanced semantic feature extraction. A Multi-scale Cascade Block (MCB) and a lightweight Channel Attention (CA) module are added between the encoding-decoding networks for optimization. In summary, this paper is well written and the experiment shows the effectiveness of the method.
Therefore, I suggests that this manuscript can be accepted after minor revision.
Detail comments:
1. It would be better if the authors highlight the novelty of this work.
2. I suggest the authors add some explanations about the main difference between this work and existing methods?
3. The section of conclusion short. I suggest the authors provide some discussions on the future work.
Author Response
Response to Reviewer 1 Comments
Thank you very much for your suggestions, we made some changes and responses as following:
Points 1: It would be better if the authors highlight the novelty of this work.
Response 1: Thank you very much for your suggestion. We have refined the description of the contribution of this work to highlight its novelty in abstract and section 1 of revised manuscript using the “Track Changes”.
Points2: I suggest the authors add some explanations about the main difference between this work and existing methods?
Response 2: Thank you for this suggestion. We have added several explanations to present the difference between this work and typical methods in abstract and section 1 of revised manuscript using the “Track Changes”.
Points3: The section of conclusion short. I suggest the authors provide some discussions on the future work.
Response 3: Thank you for pointing this out, we have added some discussion of future work in section 5 of revised manuscript using the “Track Changes”.
Thank you very much for your helpful advice, we have improved it in our paper in response to your comments, thanks again.
Reviewer 2 Report
The authors have presented a good idea and deep learning model for saliency detection. I’ve the following major comments.
1. Authors have used very limited number of metrics to compute and compare their methods’ performance against state-of-the-art. For instance, while comparing performance against SOTA, researchers use Structure Measure (S-measure), E-measure, Weighted F-measure (not the Max F-measure).
2. The authors should include some detailed information about the objective function of their method, i.e., why the authors used only the simple loss function in their method? Did they try some other objective functions as well?
3. If possible, the authors should publicize their code in a confidential platform only for the review purposes.
4. I can feel deficiency of comparison with many recent SOTA techniques, particularly, the ones based on vision transformers, such as the following:
a. VST >> Nian Liu, Ni Zhang, Kaiyuan Wan, Ling Shao, and Junwei Han. Visual saliency transformer. In Proceedings of the IEEE/CVF International Conference on Computer Vision, pages 4722–4732, 2021.
b. PASNet >> Hussain, Tanveer, et al. "Pyramidal Attention for Saliency Detection." arXiv preprint arXiv:2204.06788 (2022).
Some other representative works that authors should compete are as follows:
a. NLDF >> Ao Luo, Xin Li, Fan Yang, Zhicheng Jiao, Hong Cheng, and Siwei Lyu. Cascade graph neural networks for rgb-d salient object detection. In ECCV, pages 346–364. Springer, 2020.
b. DSS >> Qibin Hou, Ming-Ming Cheng, Xiaowei Hu, Ali Borji, Zhuowen Tu, and Philip HS Torr. Deeply supervised salient object detection with short connections. In CVPR, pages 3203–3212, 2017
c. PiCANet >> Nian Liu, Junwei Han, and Ming-Hsuan Yang. Picanet: Learning pixel-wise contextual attention for saliency detec[1]tion. In CVPR, pages 3089–3098, 2018
d. AFNet >> Mengyang Feng, Huchuan Lu, and Errui Ding. Attentive feedback network for boundary-aware salient object detec[1]tion. In CVPR, pages 1623–1632
e. PoolNet >> Jiang-Jiang Liu, Qibin Hou, Ming-Ming Cheng, Jiashi Feng, and Jianmin Jiang. A simple pooling-based design for real[1]time salient object detection. In CVPR, pages 3917–3926, 2019.
f. CSNet [14] Shang-Hua Gao, Yong-Qiang Tan, Ming-Ming Cheng, Chengze Lu, Yunpeng Chen, and Shuicheng Yan. Highly efficient salient object detection with 100k parameters. In ECCV, pages 702–721. Springer, 2020.
Author Response
Response to Reviewer 2 Comments
Thank you very much for your suggestion, we made some changes and responses as following:
Points 1: Authors have used very limited number of metrics to compute and compare their methods’ performance against state-of-the-art. For instance, while comparing performance against SOTA, researchers use Structure Measure (S-measure), E-measure, Weighted F-measure (not the Max F-measure).
Response 1: Thank you for pointing this out, we have added Weighted F-measure to further compare our work with other method’s performance in table 2 and section 4.3 of the revised manuscript using the “Track Changes”.
Points 2: The authors should include some detailed information about the objective function of their method, i.e., why the authors used only the simple loss function in their method? Did they try some other objective functions as well?
Response 2: Thank you for this constructive and useful comment. We have added explanation of why we chose this function in section 3.2 of the revised manuscript using the “Track Changes”. During our experiments, we have also tried some other objective functions as well, and the adopted one in the manuscript is optimal for our model.
Points 3: If possible, the authors should publicize their code in a confidential platform only for the review purposes.
Response 3: Thank you for this suggestion. After the paper is accepted, we will submit our code to an open source, e.g. GitHub.
Points 4: I can feel deficiency of comparison with many recent SOTA techniques, particularly, the ones based on vision transformers, such as the following:
……
Response 4: Thank you very much for this useful advice. We have added a portion of the works from the papers you provided as comparison, presented in Table 2 in the revised manuscript. The rest of the literatures are not comparable due to some reasons, such as the different datasets of the experiments. We have also newly added all the papers to the references [48]-[54] and elaborated in section 2.2 in the revised manuscript.
In addition, limited by the experimental equipment, our framework is designed based on Convolutional Neural Network (CNN), which does have deficiency compared to the transformer-based methods. Therefore, exploring transformer-based saliency methods is also a future research direction for us.
Finally, we would like to express our appreciation again to you for taking the time for a very careful reading of our manuscript, and for the insightful comments.